# Diversity of Avian Species in Peri-Urban Landscapes Surrounding Fez in Morocco: Species Richness, Breeding Populations, and Evaluation of Menacing Factors

Wafae Squalli [1,*], Ismail Mansouri [1], Ikram Douini [2], Hamid Achiban [3], Fatima Fadil [1], Mohamed Dakki [4] and Michael Wink [5]

1   Laboratory of Functional Ecology and Environmental Engineering, Faculty of Sciences and Technology, Sidi Mohamed Ben Abdellah University (USMBA), Fez 30050, Morocco
2   Laboratory of Biotechnologies and Valorization of Plant Genetic Resources, Faculty of Sciences and Technology, Sultan Moulay Slimane University of Beni Mellal, Beni Mellal 23000, Morocco
3   Laboratory of Geo-Environmental Analysis and Sustainable Development Planning, Sidi Mohamed Ben Abdelah University (USMBA), Fez 30050, Morocco
4   Laboratory of Go-Biodiversity and Naturel Patrimony, Scientific Institute, Mohammed V University, Av. Ibn Battota, 10 BP 703, Rabat 10000, Morocco
5   Institute of Pharmacy and Molecular Biotechnology, Heidelberg University, 69120 Heidelberg, Germany
*   Correspondence: wafaesqualli7@gmail.com

**Abstract:** In this study, we investigated the avian diversity and threatening factors in five peri-urban sites around Fez city (Morocco) for 2 years (2018–2019). The study hosted 131 avian species, including 64.88% breeding species, 19.84% migrant winterers, and 11.45% migrant breeders. Five species of conservation concern such as the vulnerable European turtle dove and the European goldfinch, the near-threatened ferruginous duck and bar-tailed godwit, and the endangered white-headed duck were recorded. Most bird species were recorded at the Oued Fez River (26.89%) and the El Mehraz dam (25%), followed by the El Gaada dam (17.4%), the Ain Bida garbage dump (15.5%), and the Ain Chkef Forest (15.18%). About 44.44% of the breeding species were found at Oued Fez, along with 33.33% at the El Mehraz dam, while El Gâada, Ain Chkef, and Ain Bida hosted only 7.40% of species. An important breeding population of the endangered white-headed duck was recorded at El Mehraz and Oued Fez. The extension of farmlands, urbanization, touristic activities, and drought constitute the most menacing factors for the avian diversity and their habitats in Fez.

**Keywords:** avian diversity; peri-urban area; habitats; threatening factors; Fez city

## 1. Introduction

Currently, over 50% of the human population lives in urban areas, resulting in a fast expansion of urban and peri-urban structures around the world [1]. In many cases, the rapid growth of urban landscapes has a critical negative impact on biodiversity, because many cities are built in zones of high biodiversity [2,3]. Despite the recognition of the importance of urban and peri-urban biodiversity, a general synthesis of urban biodiversity is lacking [4], mainly in developing countries, where the awareness of this problem is less developed [5].

It has been frequently claimed that peri-urban areas, including reservoirs, greenbelts, and remaining natural habitats, support a substantial avian species assemblage [6,7]. For instance, inside the administrative boundaries of Prague city, 127 of the 199 avian species that breed in the Czech Republic can be found [8]. However, this diversity is impacted by anthropogenic factors, including urbanization, pollution, traffic, degradation of natural ecosystems, noise, and disturbances [9,10]. While these features are reasonably well studied north of the Mediterranean, data are not yet available from North Africa, despite the importance of this area for both local and migratory birds [10–12].

Morocco is placed in the Mediterranean basin hotspot, one of the 25 biologically richest and most threatened ecoregions of the Earth [13]. Consequently, Morocco houses the second largest animal and vegetation diversity in the Mediterranean zone and the largest marine biodiversity [14]. More than 31,000 living species, of which about 11% are endemic, have been recorded in Morocco [15,16]. In terms of avifauna, more than 500 bird species, including breeding, wintering, and migrant species, are known from Morocco [17–20]. This bird diversity is a result of the diversity of Moroccan ecosystems [14,21], climate conditions [22], and geographical position. Morocco holds 38 RAM-SAR wetlands, 10 national/natural parks, with three sites having marine areas within their limits, and 160 sites of biological and ecological interest (SBEIs) [23]. In addition, the climate is influenced by the Atlantic Ocean (western coast), the Mediterranean Sea (northern zone), and the Sahara in the south, which together form a unique environment [24]. Moreover, Morocco is a gateway between Europe and Africa for many migrant avian species [25]. Morocco offers suitable habitats and foraging resources for many birds. However, despite this diversity and abundance, many Moroccan habitats and ecosystems are not well explored; a number of them are under human pressure, and most are already moderately to severely deteriorated. Therefore, there is an urgent need to investigate the environmental situation in Morocco.

This study had three main objectives: (1) exploration of the avian diversity (richness and abundance) among habitats surrounding Fez city in the Saiss plain; (2) assessment of breeding populations (nests, clutches, and chicks); (3) analysis and mapping of most threatening factors to avian species in each habitat. We selected the Fez region because of its central location in Morocco, which is close to the humid Atlas Mountains, offering a wide range of ecosystems including forests, reservoirs, and farmlands.

The study area offers a last foraging opportunity for long-distance migratory birds before they cross the Sahara [26]. Equally, the region holds several wetlands, considered as important RAMSAR sites for water birds. Generally, this study was designed to explore the importance of habitats for birds and to characterize the most threatening factors to both habitats and birds.

## 2. Materials and Methods

### 2.1. Study Area

The city of Fez is considered the second largest city in Morocco, hosting a population of nearly three million inhabitants. It is located in the center of Morocco, in the Saiss plain situated between the Rif Mountains (North) and the Atlas Mountains (South) (Figure 1) [27,28]. The climate of the area is characterized by cold winters and hot and dry summers. The diurnal temperature varies from 17 to 34 °C in July, and from 4 to 15 °C in January. The rainfall is higher in the winter period, whereas summers are almost completely dry [29]. Average rainfall in the Saiss plain is 500 mm/year, enabling rainfed agriculture of cereals and forage crops over large areas. Irrigated agricultural cultivation, principally horticulture and fruit trees, essentially depends on groundwater. According to the Ministry of Agriculture and Fisheries statistics, in 2012, about 45,000 ha of farmland was irrigated with groundwater in the Saiss plain, representing more than 91% of the surface in the area. In addition to agriculture, the study area is characterized by a diversity of wetlands, including dams, lakes, and rivers [30,31]. These ecosystems host an important community of birds and other animals [30,32].

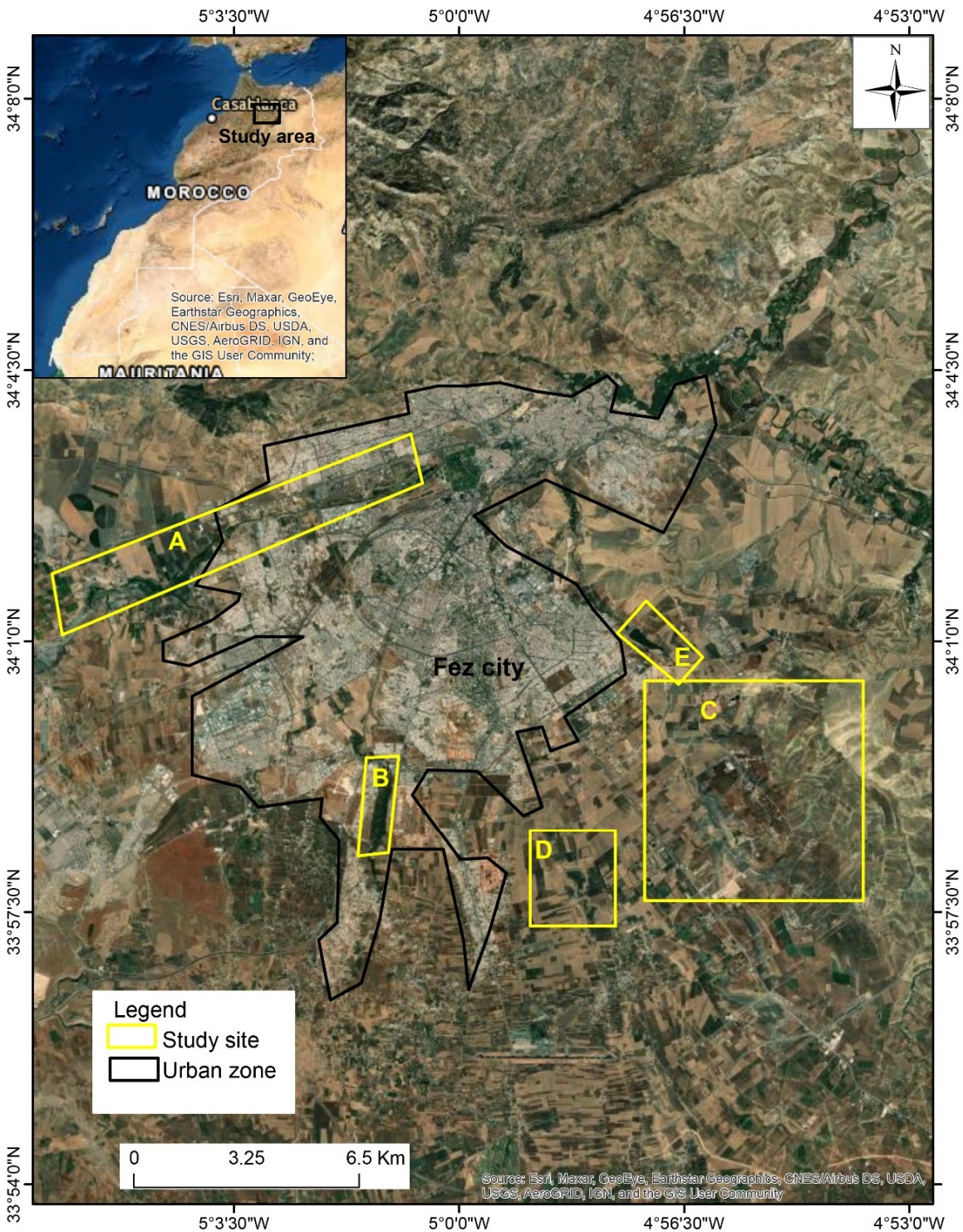

**Figure 1.** Location of Fez city and studied habitats (between 2018 and 2019). (**A**) Oued Fez river; (**B**) Ain Chkef Forest; (**C**) Ain Bida Dump; (**D**) El Mehraz Dam; (**E**) Gaada Dam.

## 2.2. Bird Survey

The study zone was divided into five principal habitats (Figure 2), including Oued Fez River, Ain Chkef Forest, Ain Bida Dump, and the El Gaada and El Mehraz dams. These sites were selected in the peri-urban areas surrounding Fez (between 15 and 20 km to the city center). These ecosystems were considered as peri-urban on the basis of the master plan of Fez city considering lands in the range of 30 km around the city as peri-urban, as

they are programmed to be urbanized in the next two decades. Equally, these ecosystems were directly influenced by the activities of the city (i.e., urbanization, wastewater, solid wastes, touristic visits, etc.). These selected sites are the dominant ecosystems in the area and represent the only systems located in the peri-urban zones between Fez city and rural farmlands of Saiss plain.

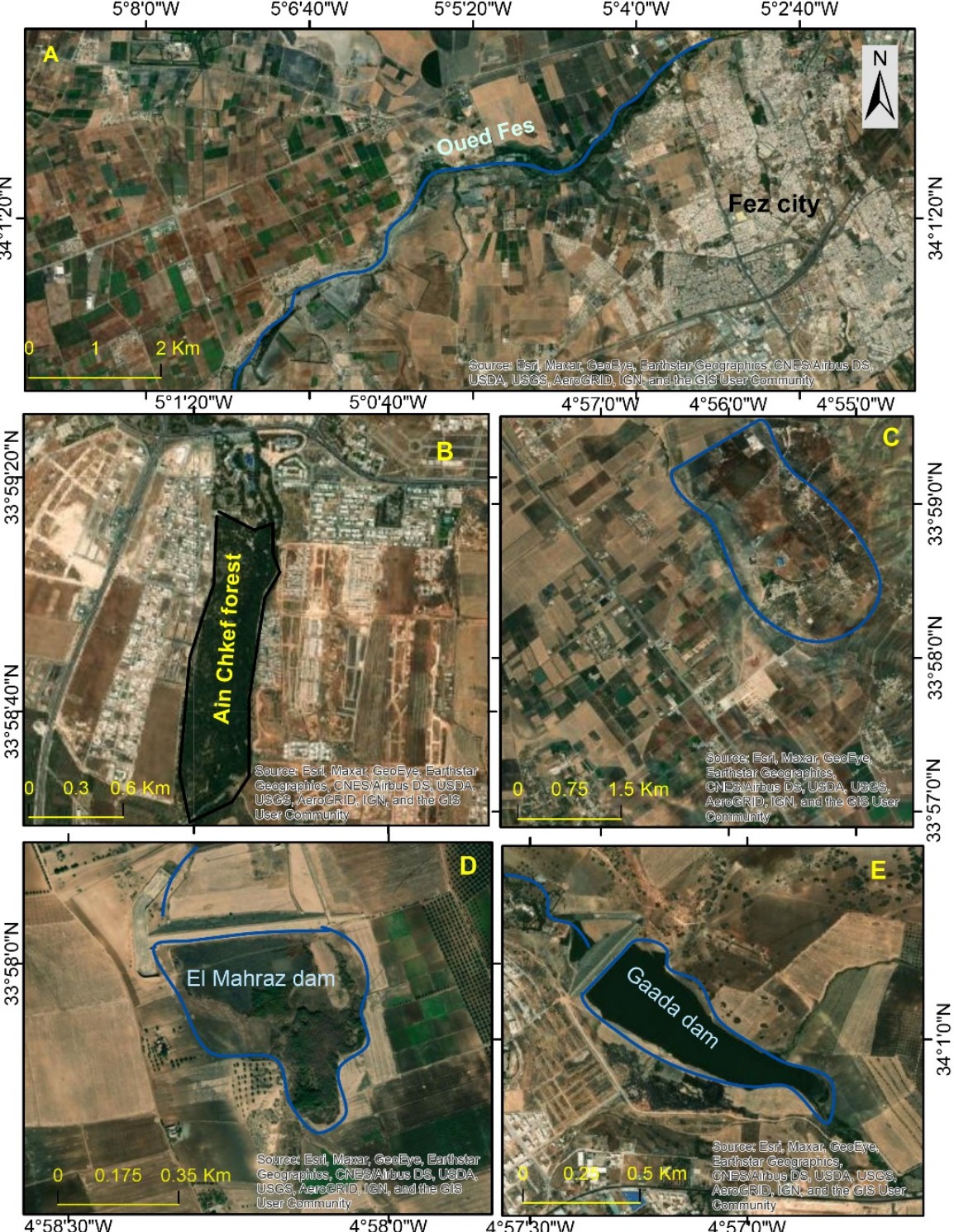

**Figure 2.** Studied sites: (**A**) Oued Fez; (**B**) Ain Chkef Forest; (**C**) Ain Bida Dump; (**D**) El Mehraz Dam; (**E**) Gaada Dam.

At each site, bird species and abundance were surveyed twice monthly and recorded during the breeding and wintering seasons from January 2018 to December 2019. At the river, garbage dump, and dams, bird species were recorded by means of point-counts with unlimited distance due to the wide area explored (i.e., 200 ha in Oued Fez river) [33]. "Point-counts" represent an effective method for assessing bird abundance [34], because they permit an extensive surveying of study plots and the neighboring landscapes [35,36]. Instead, Ain Chkef Forest was explored via line-transect (2–5 km) due to the nature and surface of the studied habitat (only 54 ha of woody forest). In each transect, observation points had a distance of 300 to 500 m from each other, and the count duration was around 15 min per point. In aquatic ecosystems and the garbage dump, data on bird species were collected by direct observations with the help of binoculars and telescopes (Olympus 8 × 40 and 10 × 50, BRAUN PHOTO TECHNIK GmbH, Nürnberg/Germany). At the El Gaada and El Mehraz dams, three observation points (20 to 30 min per point) were selected (two on the borders and one in the center) in the riparian vegetation to cover the entire ecosystem and to avoid disturbance of birds. At the Ain Bida Dump, two observation points were selected on the nearest hills surrounding the site (to get a panoramic view). The combination of these methods was aimed at collecting wide-ranging ecological data in a cost-effective manner due to the structural differences among studied sites [37]. Because of the vast surface area that these habitats cover, the higher density of aquatic birds, and the difficulty in accurately identifying some species (i.e., some birds dive underwater for extended periods of time), aquatic ecosystem surveys have lasted longer than terrestrial ones.

### 2.3. Breeding Populations

At each site, we documented breeding species using direct and indirect approaches. In direct observations, we searched nests and eggs in riparian vegetation, farming fields, and other structures surrounding aquatic ecosystems (rivers and dams) and the dump, while, in the forest of Ain Chkef, nests were searched in trees, shrubs, and riparian cover. In the indirect approach, we searched for chicks and sub-adults of each species after hatching and fledging dates. This approach was very useful for species that hide their nests, principally far from studied sites. Furthermore, we noted the number of nests, eggs, and or chicks and the observation period.

### 2.4. Environmental Variables and Mapping of Threatening Factors

In parallel with avian diversity, we recorded natural variables including vegetation cover and water body, as well as anthropogenic factors, such as tourists, cars, farmlands, sanitation tunnels, fishing activity, egg robbery, and livestock, during each visit to studied habitats (Table 1). Descriptions of the variables and methods are summarized in Table 1. These variables are suggested to influence avian diversity in direct and indirect manners. At the El Mehraz Dam, we recorded a drought period, but we did not include it in the statistical analysis (exclusive event).

For mapping threatening factors, the spectral data sources of Landsat Enhanced Thematic Mapper Plus (ETM1) images from January 2015 to December 2019 were used in the classification procedure. Generally, the Landsat (ETM1) beam records seven groups of spectral data in the infrared, visible, and thermal ranges of the electromagnetic spectrum. Multidate imagery was used to describe the dry season at the El Mehraz dam between the wet (April) and dry season (October). A total of 58,675 training pixels were used to categorize the 907,429 pixels contained in the study area. Classification training sites were established to study habitats, urbanized areas, farmlands surfaces, and other land cover using recently digitized wetland and riparian data acquired from 1:22,000 color infrared (CIR) aerial photography of the monitored zone and onsite assessments (mainly to illustrate the farming and other human activities in the El Mehraz watershed). Seven land-cover types were identified in the final classification method: urban extension, Water body, forest (natural plantations), matorrals (steppe vegetation), agriculture (fruit trees, cereals, vegetables, etc.), watershed (channels, streams, and rivers converted to the reservoirs),

andquarriers. The maps were created using QGIS 3.14 (open-source). Lastly, for greater accuracy, maps were built for the study years and extended to the last 20 years to clarify the evolution of land use around the studied habitats.

**Table 1.** Natural and anthropic factors recorded in study sites.

| Variables | Description | Methods |
|---|---|---|
| Vegetation cover | Surface covered by the vegetation surrounding each study site | Estimated in the field and by satellite images (QGIS and Google Earth Pro) |
| Water body or habitat dimension | Surface of water stream (river), water reservoirs (dams), and water pools (dump) | Estimated in the field and by satellite images (QGIS and Google Earth Pro) |
| Distance to urban center and nearest urban zone | Distance separating the habitat to the centre of Fez city | Estimated by satellite images (QGIS and Google Earth Pro) |
| Tourists | Number of persons recorded in the field during each visit | Estimated for each site per visit |
| Sanitation tunnels | Number of dump sewage tunnels leading into the studied sites | Recorded in the field (illegal ones) and from the urban master plan of Fez (legal ones) |
| Fishing activity | Fishers observed at each site | Estimated for each visit |
| Egg robbery | People or children robbing eggs of breeding birds | Number of robbed eggs and people captured in the field by regional department of water and forestry in Fez |
| Livestock | Total number of animals recorded at each site (total of all visits) | Recorded during each visit |

*2.5. Statistics*

We classified recorded species and their populations following families and orders. We also noted the phenological and conservation status of recorded birds according to the latest updates on Moroccan birds [38] and red lists of BirdLife International [39]. Biological diversity indices were calculated to compare studied sites. Various types of total species diversity indices including Shannon–Wiener species diversity index (H) [40], Margalef species richness index (D) [41], and Simpson index (D) [42,43] were calculated (considering 12 months and 2 years of monitoring).

The diversity indices (Shannon–Wiener index, Margalef index, and abundance index) were calculated and compared for all habitats. Equally, the number of species, families, and abundance were calculated and compared by means of one-way ANOVA. Bray–Curtis similarity coefficient ordination was used to compare similarity among studied habitats. To evaluate the principal factors predicting the richness of avian species (BS) in studied habitats, recorded human activities, including tourists (T), cars used by visitors (C), fishing activity (FS), egg robbery (ER), livestock (L), sanitation tunnels (SN), and natural variables such as natural vegetation cover (NVC) and surface of water body (SWB) were considered as explanatory variables, while the abundance of bird species (BS) was considered as a response variable, analyzed using PCA (only eigenvalues >1.0 were selected). Similarly, the impact of touristic visits was analyzed considering the number of persons per month (January to December) as explanatory variables. This variable is suggested to disturb breeding peers during the reproductive season. We evaluated the relationship among the number of species, their populations, and habitat characteristics (area, distance to nearest urban zone, and distance to urban center) via linear regression. Bird species and abundance were considered dependent variables, while habitat area, distance to nearest urban zone, and distance to the urban center were considered predictors. All analyses were performed using SPSS 18, and results were considered significant at $p < 0.05$.

**3. Results**

Table S1 documents the bird species diversity of the prospected sites in the Fez area. In total, 131 bird species were documented, belonging to 46 families, including Anatidae (9.16% of species), Scolopacidae (8.39% of species), Fringillidae (6.86%), Ardeidae (6.10%), Accipitridae (3.8%), and Sylviidae (2.3% of species), while Laridae (1.52%), Recurvirostri-

dae (0.76%), Pandionidae (0.76%), Phalacrocoracidae (0.76%), Threskiornithidae (0.76%), Alcedinidae (0.76%), and Troglodytidae (0.76%) were the least represented. Moreover, *Apus apus* (Apodidae) and *Sturnus unicolor* (Sturnidae) were the most abundant species.

The documented species had different phenological and conservation statuses. Specifically, 85.5% of the birds were breeding species, 8.4% were migrant birds, and 31.3% were wintering species. Five species of conservation concern included the vulnerable *Streptopelia turtur* and *Carduelis cannabina*, the near-threatened *Aythya nyroca* and *Limosa lapponica*, and the endangered *Oxyura leucocephala*. The other bird species (126) were species of less conservation concern.

### 3.1. Habitat Richness and Breeding Species

The results of diversity analysis (Margalef index, Shannon–Wiener index, and Simpson index) and compositional parameters (species richness) are shown in Table 2. Wetlands, mainly Oued Fez, and El Mehraz hosted the highest avian diversity with 164 species, while the Ain Chkef Forest, the Ain Bida Dump, and the El Gaada Dam hosted a smaller number of avian species ($n = 24$ months (2 years), $f = 10.87$, $df = 271.8$, $p < 0.001$). On the other hand, the Bray–Curtis similarity coefficient ordination indicated that Ain Bida and Ain Chkef were similar in bird community composition. The next closest similar ecosystem was Oued Fez, followed by El Mehraz. The least similar ecosystem was El Gaada, which was least similar to Ain Bida and Ain Chkef, but more similar to El Mehraz (Figure 3).

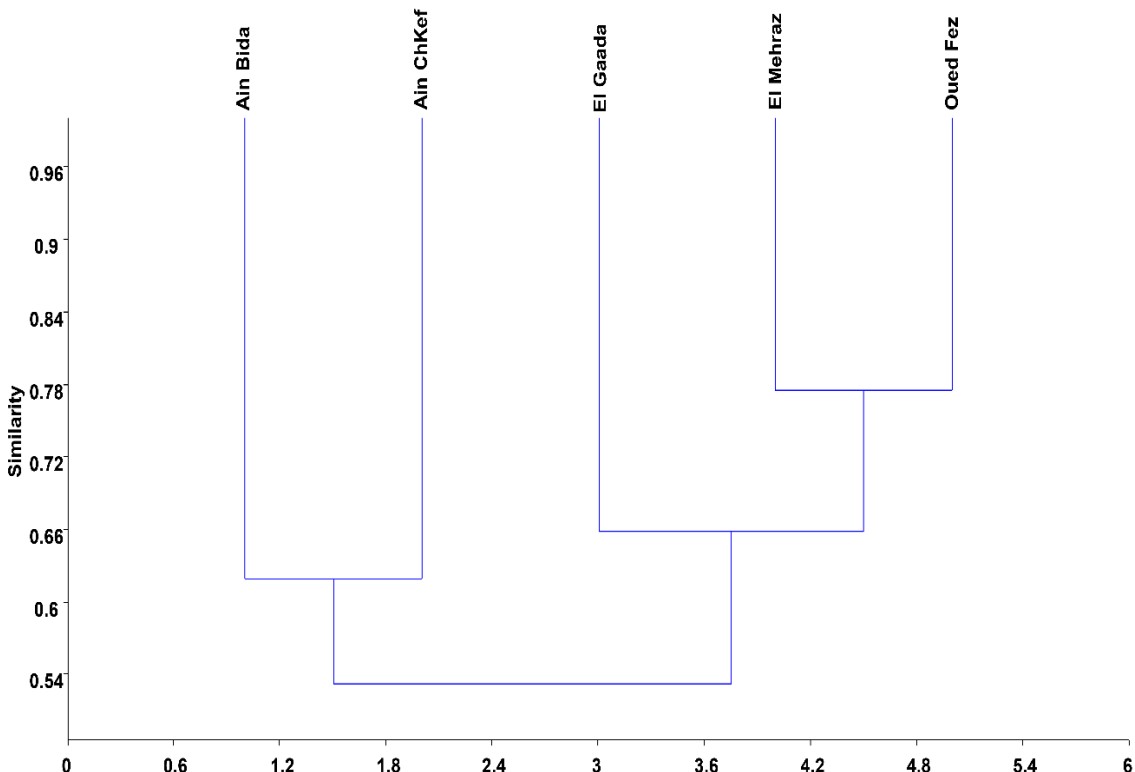

**Figure 3.** Bray–Curtis similarity coefficient ordination comparing studied sites based on avian communities.

The Fez peri-urban zone offers favorable breeding habitats for several species (Table 3). In total, 27 species were documented as breeding birds: 44.44% at Oued Fez, 33.33% at El Mehraz, 7.4% at El Gaada, 7.4% at Ain Chkef, and 7.4% at Ain Bida. Moreover, three species with conservation concern were recorded as breeders, namely, the vulnerable *Streptopelia turtur arenicola* (120 nests at both Ain Bida and El Mehraz), the endangered *Oxyura leucocephala* (36 nestlings at both Oued Fez and El Mehraz), and the near-threatened *Aythya nyroca* (four chicks at El Mehraz).

**Table 2.** Avian diversity among studied habitats in Fez peri-urban area between 2018 and 2019.

| | Oued Fez | El Mehraz | El Gaada | Ain Bida | Ain Chkef |
|---|---|---|---|---|---|
| Number of taxa | 85 | 79 | 55 | 49 | 48 |
| Shannon–Wiener | 4.443 | 4.216 | 3.799 | 3.892 | 3.871 |
| Simpson | 0.9882 | 0.977 | 0.9624 | 0.9796 | 0.9792 |
| Margalef | 18.91 | 17.42 | 12.98 | 12.33 | 12.14 |

**Table 3.** Breeding populations of avian species recorded in Fez peri-urban area between 2018 and 2019.

| Species | Nests | Eggs | Chicks | Breeding Periods | Breeding Site |
|---|---|---|---|---|---|
| *Oxyura leucocephala* | - | - | 4<br>32 | June–July<br>June–July | Oued Fez<br>El Mehraz |
| *Aythya nyroca* | - | - | 4 | June–July | El Mehraz |
| *Streptopelia turtur* | 62<br>58 | 96<br>90 | 70<br>63 | April–August | Ain Bida<br>El Mehraz |
| *Himantopus himantopus* | 31 | >168 | 140 | April–July | Oued Fez |
| *Fulica atra* | 131 | >322 | 230 | February–July | Oued Fez<br>El Mehraz |
| *Fulica cristata* | 86 | >251 | 128 | February–July | Oued Fez<br>El Mehraz |
| *Nycticorax nycticorax* | 6 | 26 | 20 | May–June | Oued Fez |
| *Ardeola ralloides* | - | - | 14 | July–August | Oued Fez |
| *Tachybaptus ruficollis* | 3 | - | 36 | April–August | Oued Fez<br>El Mehraz |
| *Podiceps cristatus* | - | - | 16<br>4 | March–August | EL Gaada<br>Oued Fez |
| *Anas platyrhynchos* | 53 | >231 | 84<br>22<br>43<br>14 | March–August | Oued Fez<br>El Gaada<br>El Mehraz<br>Ain Chkef |
| *Aythya ferina* | 5 | - | 12<br>31 | March–August | El Mehraz<br>Oued Fez |
| *Porphyrio porphyrio* | 2 | - | 18 | March–August | Oued Fez |
| *Gallinula chloropus* | 2 | - | 58 | March–August | Oued Fez |
| *Pica mauritanica* | 25 | - | 7 | March–August | Ain Bida |
| *Accipiter nisus* | 1 | - | 3 | April–August | Ain Chkef |
| *Circus aeruginosus* | 1 | 5 | 7<br>6 | March–August | El Mehraz<br>Oued Fez |
| *Carduelis carduelis* | -<br>-<br>- | -<br>-<br>- | 15<br>18<br>7 | April–August | El Mehraz<br>Ain Bida<br>Oued Fez |
| *Fringilla coelebs* | 13 | 21 | 12 | April–August | El Mehraz<br>Ain Bida |
| *Chloris chloris* | 8 | 28 | 13 | April–August | El Mehraz<br>Ain Bida |
| *Serinus serinus* | 36 | 56 | 41 | April–August | El Mehraz<br>Ain Bida |
| *Passer domesticus* | 23 | 43 | 80 | March–August | Oued Fez |
| *Turdus merula* | 11 | 17 | 12 | April–August | El Mehraz<br>Ain Bida |
| *Hirundo rustica* | 4 | - | - | April–July | Oued Fez |
| *Delichon urbicum* | 1 | 0 | - | April–July | Oued Fez |
| *Motacilla alba* | - | - | 8 | April–July | El Mehraz<br>Ain Bida<br>Oued Fez |
| *Motacilla cinerea* | - | - | 26 | April–July | El Mehraz<br>Oued Fez |

### 3.2. Predicting and Threatening Factors

The relationships between the diversity and abundance of avian species and the potential predictors of habitats including area (ha), distance to the nearest urban zone, and distance to the urban center (km) are summarized in Table 4 and Figure 3. Results show that the diversity of avian species and their abundances were not related to the area of habitat or distance separating sites from nearest urban zone and urban center.

**Table 4.** Predictors (area, distance to urban center, and distance to nearest urban zone) of diversity (number of species) and abundance of avian species in studied habitats.

| Dependent Variables | Predictors | Sum of Squares | Df | Mean Square | F-Ratio | *p*-Value |
|---|---|---|---|---|---|---|
| Species | Area | 442.832 | 1 | 442.832 | 1.7 | 0.2834 |
| | Residual | 781.968 | 3 | 260.656 | | |
| | Distance to urban center | 137.734 | 1 | 137.734 | 0.38 | 0.5812 |
| | Residual | 1087.07 | 3 | 362.355 | | |
| | Distance to nearest urban zone | 10.8335 | 1 | 10.8335 | 0.03 | 0.8804 |
| | Residual | 1213.97 | 3 | 404.655 | | |
| Populations | Area | $8.92 \times 10^9$ | 1 | $8.92 \times 10^9$ | 0.36 | 0.5929 |
| | Residual | $7.53 \times 10$ | 3 | $2.51 \times 10$ | | |
| | Distance to urban center | $2.00 \times 10$ | 1 | $2.00 \times 10$ | 0.94 | 0.4046 |
| | Residual | $6.42 \times 10$ | 3 | $2.14 \times 10$ | | |
| | Distance to nearest urban zone | $4.21 \times 10$ | 1 | $4.21 \times 10$ | 3 | 0.1819 |
| | Residual | $4.21 \times 10$ | 3 | $1.40 \times 10$ | | |

Oued Fez River and its avian populations were impacted by fishing activities, egg robbery, and abundant sanitation tunnels (Figure 4). Visitors and their cars were the most important disturbing factors in the Ain Chkef Forest. Ain Bida Dump and El Gaada Dam were impacted by a huge concentration of livestock.

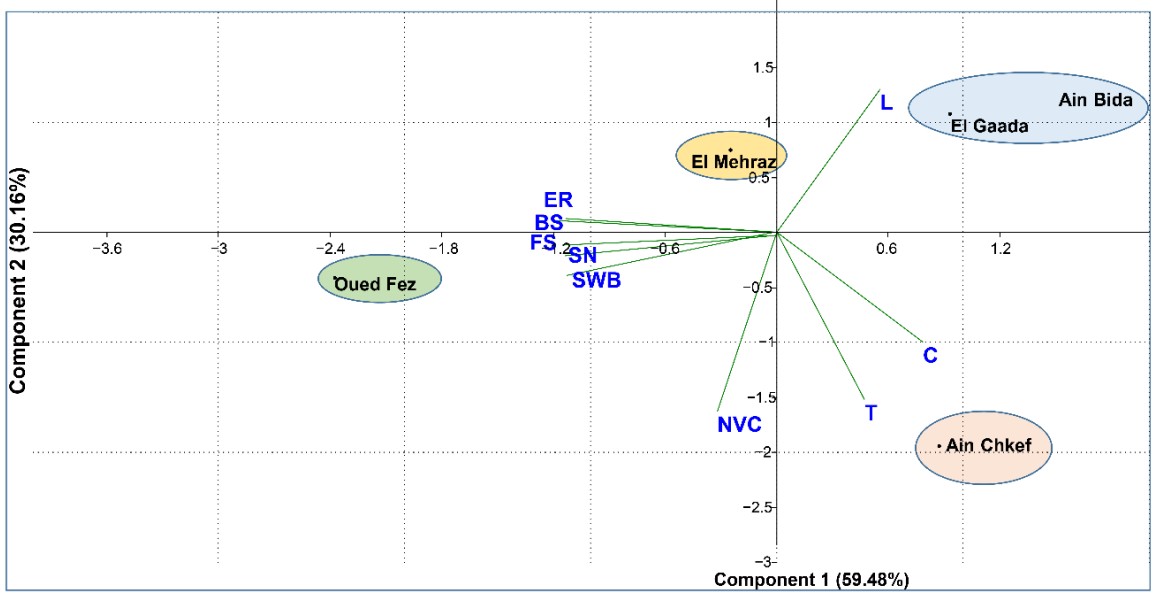

**Figure 4.** Summary of principal factors disturbing the avian diversity and their habitats in Fez peri-urban area (BS: bird species, C: cars, ER: egg robbery, FS: fishing activity, L: livestock, NVC: natural vegetation cover, SN: sanitation tunnels, SWB: surface of water body, T: tourists).

The Fez urban and agricultural surfaces have grown over the last 20 years (Figure 5). The urban area of Fez covers 37% of the studied zone. Furthermore, urban settlements have encircled 100% of the Ain Chkef Forest, 92% of the El Gaada Dam, 59.8% of the Oued Fez River, and 32.3% of the El Mehraz Dam. In parallel, the farmlands (mainly olives and cereals) have covered 100% of the Ain Bida Dump, 63.5% of the El Mehraz Dam, 43.7% of the Oued Fez River, and 7% of the El Gaada Dam.

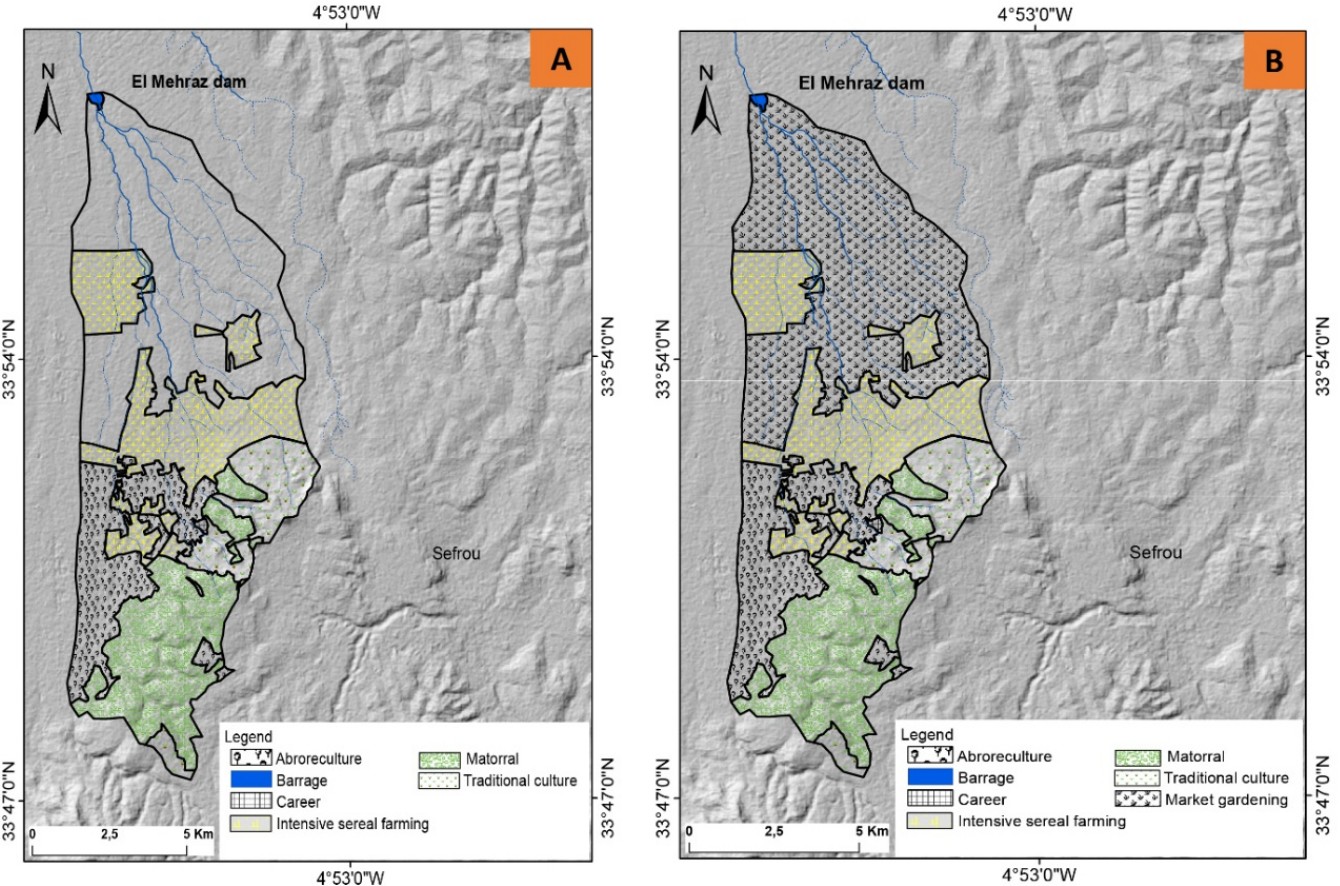

**Figure 5.** The expansion of farmland and urban landscapes in Fez peri-urban areas over the last 20 years: (**A**) watershed farmlands in 2020; (**B**) watershed farmlands in 2020.

Studied habitats were frequented by a substantial number of tourists. The Ain Chkef Forest and El Gaada Dam were visited by a total of 341,828 and 70,179 tourists, respectively, principally during the October–January and March–June periods (Figure 6). The El Mehraz Dam (1049 visitors), the Ain Bida Dump (3958 visitors), and the Oued Fez River (18,652 visitors) were visited by a lower number of tourists.

During 2017–2018 and 2019–2020, the El Mehraz dam was completely dry (Figure 7). The populations of water birds (79 species), including resident and migrant species, were directly impacted. Moreover, breeding attempts of the white-headed duck (32 nestlings) and the ferruginous duck (four chicks) were not successful due to the absence of water and feeding resources. The main factor influencing the water body of the dam was represented by farmlands (olives and vegetables) that require huge quantities of water for irrigation, particularly during the hot summer, coinciding with the breeding period of the majority of recorded waterbirds.

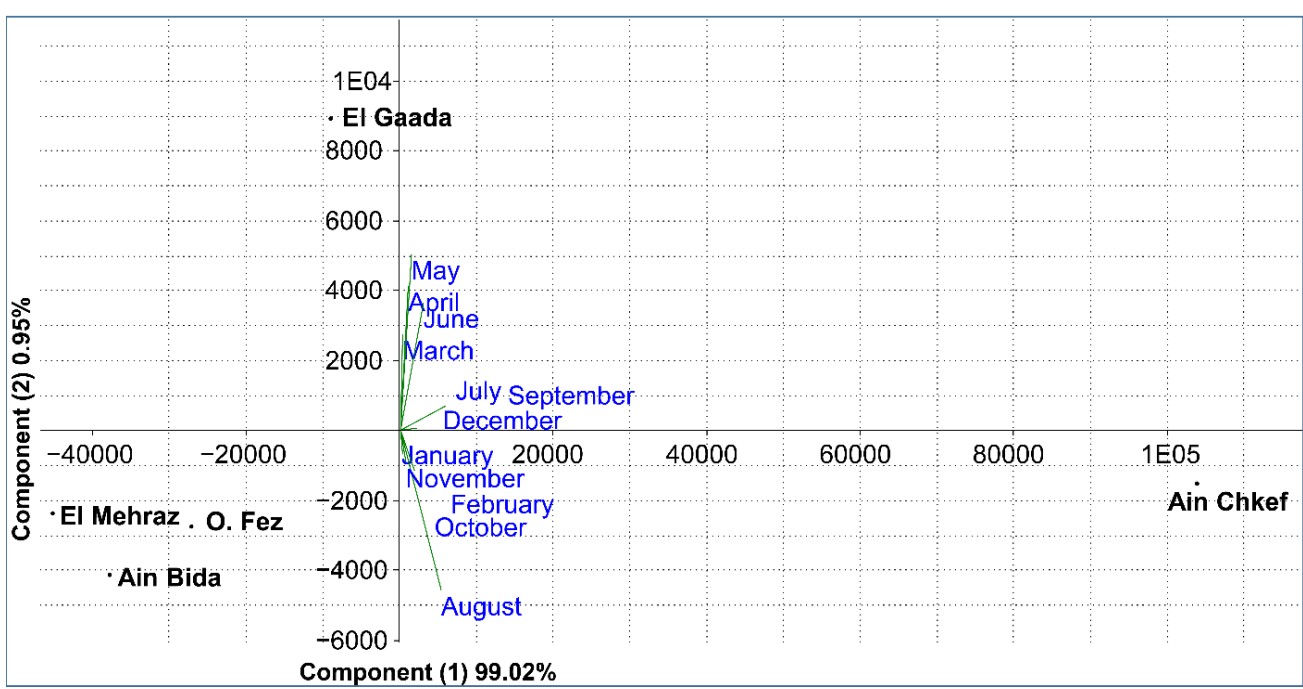

**Figure 6.** Periods of tourists visiting studied sites between 2018 and 2019.

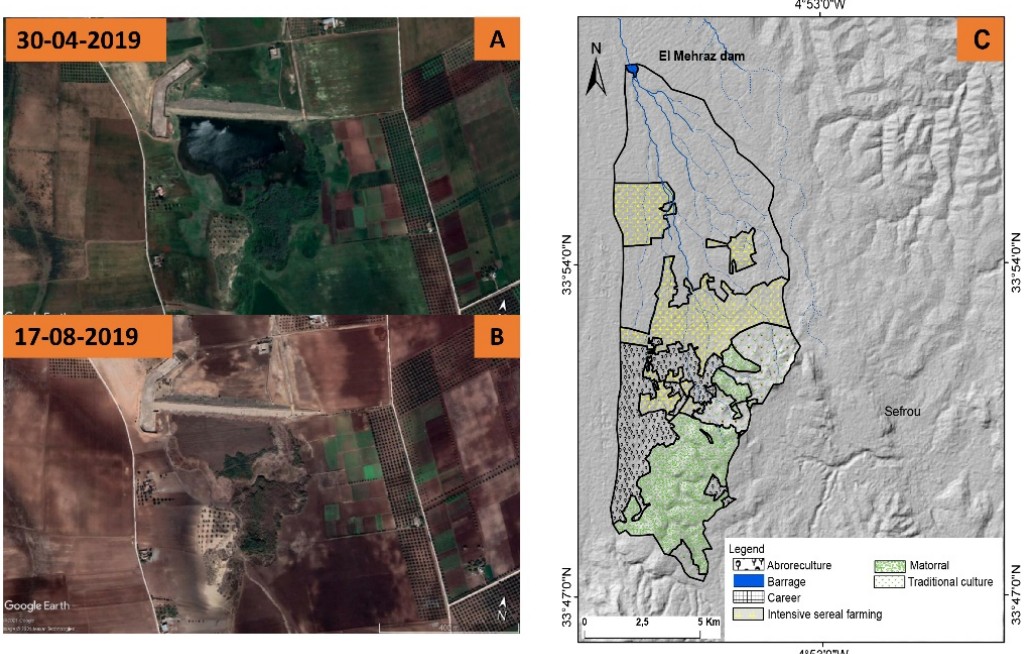

**Figure 7.** The impact of farmlands developed in the El Mehraz watershed: (**A**) availability of water in the dam reservoir; (**B**) dry reservoir, and (**C**) watershed farmlands.

## 4. Discussion

According to a thorough literature research, this is the first systematic study of the avian diversity in peri-urban areas in Morocco, which is important for western Palearctic migrants [26,44–47]. Our main objective was to deliver solid data on avian diversity in the ecosystems surrounding the historical city of Fez. We obtained new and valuable data describing bird diversity, habitat use, and threatening factors, important for future monitoring and long-term conservation activities of the threatened avian species and populations.

In the peri-urban zone of Fez, we recorded a total of 131 bird species belonging to 41 families and 17 orders. Apparently, this avian richness is the highest in all peri-urban

areas of North Africa. In urban areas of Midelt (180 km south of Fez, Morocco) only 33 bird species were recorded between 2015 and 2019 [26]; in the urban area of Ain Bida (region of Oum El Bouaghi, Algeria), only 33 birds were observed between 2013 and 2014 [48]; in Gabes (Southeastern Tunisia), only 43 birds were located around the urbanized perimeter [49]. In our case, Accipitridae, Muscicapidae, and Alaudidae represented 34% of recorded species. Equally, the orders of Passeriformes and Gruiformes were the most observed in the area. This is in agreement with results cited for other Moroccan regions; in Midelt, Accipitridae, Muscicapidae, Alaudidae, and Anatidae represented 29% of avian species [26]; in the Middle Atlas (180 km to South of Fez), Accipitridae, Muscicapidae, Alaudidae, and Motacillidae represented 36.33% of birds [50]; in High Atlas (450 km to West of Fez), Muscicapidae, Fringillidae, and Accipitridae represented 30.43% of total species [51]. On the other hand, five species of conservation concern, namely, the vulnerable turtle dove and the European goldfinch [52], the near-threatened ferruginous duck and bar-tailed godwit [53], and the endangered white-headed duck [54] were observed regularly (except the bar-tailed godwit, which was recorded as a wintering species) in the area, and this necessitates important and urgent attention to characterize their habitats for effective conservation measures [55,56]. The avian diversity was higher in aquatic ecosystems such as the Oued Fez River, El Mehraz Dam, and El Gaada Dam, while, in terrestrial habitats, such as the forest of Ain Chkef and Ain Bida Dump, the number of species was lower. Generally, the diversity of birds is higher in wetlands because of their richness in water, food abundance (i.e., fish, algae, insects, and fruits), and vegetation cover [57–59], while, in terrestrial habitats, a low availability of water limits the presence of birds [60–62]. Similar results were cited in Midelt, where 91 birds were recorded in wetlands compared to 69 species in forests and 13 birds in landfills [26].

Among recorded birds, 27 avian species were documented as breeding birds, representing 20.61% of all recorded species. Most breeders were concentrated in the Oued Fez River and at the El Gaada Dam as compared to forests and the dump. This was seemingly due to the abundance of supporting trees (vegetation cover) and foraging elements during breeding seasons in wetlands [63]. Moreover, three species of conservation concern were recorded as breeders, namely, the vulnerable *Streptopelia turtur arenicola* at both Ain Bida and El Mehraz, the endangered *Oxyura leucocephala* at both Oued Fez and El Mehraz, and the near-threatened *Aythya nyroca* at El Mehraz dam. Similarly, Squalli et al. (2021, 2022) [32,64] revealed a successful breeding attempt of *Streptopelia turtur arenicola* in olive groves surrounding Fez city, while Ouassou et al. (2018) [30] reported only a substantial wintering population of *Oxyura leucocephala* at the Oued Fez and El Mehraz Dam. However, these breeding cases are a good indicator for the importance of the area for rare and endangered species [30,65], particularly for those that have lost their natural habitats in other Mediterranean zones, such as *Oxyura leucocephala* and *Fulica cristata* that have lost their breeding habitats in south of Spain [66,67], and *Aythya nyroca* that is threatened in Algeria [68]. Therefore, these habitats need more advanced studies to delimit nesting sites, as well as to evaluate the rates of breeding success and available foraging resources of these threatened species and their broods.

The avian species and their habitats in the Fez peri-urban zone were threatened by anthropogenic and natural factors, similar to other North African and Mediterranean ecosystems [69–72]. In this study, urbanization extended to the natural habitats of the Oued Fez river and El Gaada, while the extension of farmlands reduced the surfaces of natural habitats and increased the water exploitation for irrigation, which reduced water bodies for aquatic birds. In the summers of 2018 and 2019, the entire water reservoir of the El Mehraz Dam was low, due to low precipitation and extensive water pumping for irrigation of surrounding farmlands. As a consequence of the drought, the breeding of the whole population of *Oxyura leucocephala* and *Aythya nyroca* failed. This situation is expected to be aggravated in the near future because of the semiarid status of the area [29,73] and the impacts of climate change with increased temperatures and reduced precipitation regimes [74,75]. Equally, we recorded a substantial impact of tourists and fishers in studied

habitats, which increased the disturbance of the avian population, while egg robbery impacted the breeding success of nesting birds.

## 5. Conclusions

In summary, this investigation provided a first detailed investigation of the avifauna and habitat use in peri-urban areas of Fez in Morocco, important for Palearctic migrants. The most menacing factors for birds were documented. These results could be of great interest for future long-term monitoring and conservation measures, mainly to protect habitats and the most endangered populations. We stress the importance of water supply and vegetation cover for conservation to provide birds with suitable and safe nesting sites, as well as sufficient foraging resources.

**Supplementary Materials:** The following supporting information can be downloaded at: https://www.mdpi.com/article/10.3390/d14110945/s1, Table S1: (Abundance 2018–2019), phenological status (acc: accidental, b: breeding, m: migration, r: resident, w: wintering, s: summering) and IUCN conservation status (E: Endangered, LC: Least Concern, NT: Near threatened, VU: Vulnerable) of recorded species in Fez pre-urban zone.

**Author Contributions:** Conceptualization, W.S. and I.M.; methodology, validation, and formal analysis, W.S., I.M., I.D. and H.A.; writing—original draft preparation, W.S.; writing—review and editing, F.F. and M.D.; visualization, M.W. All authors have read and agreed to the published version of the manuscript.

**Funding:** This research received no external funding.

**Institutional Review Board Statement:** Not applicable.

**Data Availability Statement:** The data used to support the findings of this study are included within the article.

**Conflicts of Interest:** M. Wink is the Editor-in-Chief of Diversity.

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
