# Peer review of "Diversity of Avian Species in Peri-Urban Landscapes Surrounding Fez in Morocco: Species Richness, Breeding Populations, and Evaluation of Menacing Factors"

_diversity, doi:10.3390/d14110945_

Round 1

Reviewer 1 Report (Previous Reviewer 2)

After the first review the authors considered nearly all of my comments

There are only a few things to criticise.

My main concern is the combination of different survey methods to calculate one population size at the end. This should be addressed at least in the method section. I am also concerned about the wide range in population size in the supplemental material. Did you really count 40.000 Columbia livia or is an estimation, especially because of 10.000 to 40.000.

I would call Table SM 1.as Appendix and for

Streptopelia decaocto                   r/b         LC           20000-3000

there Is a 0 to much or to less.

I still miss a definition of Acc: accidental, b: breeding, m: migration, r: resident, w: wintering, S: summering – all abbreviation should be in small or large letters

Table 1 – find a better format, size of letters, spacing of lines

Table 2 – use Simpson and Margalef instead of Simpson_1-D and Margalef-D

What is the difference between table 3 and Table 1 SM?

Both refer to Fez pre-urban area. Table 3 is breeding population and Table 1 SM is the population all year round?

Table 3 – is - April-August correct or is it April – August? There are several lines with this

Table 4: slightly edit the format with no deviation in the line

All in all the paper is well investigated and written. has good tension and describes the problem were well.

Author Response

After the first review, the authors considered nearly all of my comments

There are only a few things to criticize.

My main concern is the combination of different survey methods to calculate one population size at the end. This should be addressed at least in the method section. I am also concerned about the wide range of population size in the supplemental material. Did you really count 40.000 Columbia livia or is an estimation, especially because of 10.000 to 40.000.

Response: Thank you for your interest. The combination between transects and fixed point is due to the difference between features of studied habitats.

  • To study birds in forests and farmlands, ornithologists use transects and point of count as we did: see studies below:

Mohamed Mounira, Mohamed Dakkib, Ikram Douinia, El-Mostafa Benkaa, Ouibimah Abdessamada, Ayoub Nouria, Ismail Mansouric, Soumaya Hammada. The avifauna of two High Atlas valleys: breeding assemblages in forest stands and open lands. J. Anim. Behav. Biometeorol., vol.10, n3, 2225, 2022. http://dx.doi.org/10.31893/jabb.22025

Fujita, M.S., Samejima, H., Haryadi, D.S. et al. Low conservation value of converted habitat for avifauna in tropical peatland on Sumatra, Indonesia. Ecol Res 31, 275–285 (2016). https://doi.org/10.1007/s11284-016-1334-2.

  • To estimate populations of waterbirds in wetlands (lakes, dames, and reservoires), scientist use observation points withouth transects as we did (the number of points depends on the surface couvered by the habitat: see below:

Maclean, I.M.D., Bird, J.P. & Hassall, M. Papyrus swamp drainage and the conservation status of their avifauna. Wetlands Ecol Manage 22, 115–127 (2014). https://doi.org/10.1007/s11273-013-9292-8

Equally, to observe avian species in farmlands and forests, you need transect lines to increase the detection probability, while aquatic berds are relatively stable which necessitate fixed points in camouflaged sites.  

For the number of Columbia It’s a 4000-10000 ans it’s a really count not estimation. I correct the number in the table.

I would call Table SM 1.as Appendix and for

Streptopelia decaocto                   r/b         LC           20000-3000

there Is a 0 to much or to less.

Response: Its 2000-3000

I still miss a definition of Acc: accidental, b: breeding, m: migration, r: resident, w: wintering, S: summering – all abbreviation should be in small or large letters

Response: Accidental: Birds that show up outside of their normal range, Route of migration or season

Breeding: birds that nest and breed in Morocco,

Migration: birds that migrate between Morocco and other regions in Africa or Europe

Resident: Birds with important local populations in Morocco,

Wintering: Birds that pass wintering periods in Morocco,

Summering: Birds that pass summer and spring in Morocco,

Table 1 – find a better format, size of letters, spacing of lines

Response: Done

Table 2 – use Simpson and Margalef instead of Simpson_1-D and Margalef-D

Response: done

What is the difference between table 3 and Table 1 SM?

Response: Table 1 SM records all species that have been seen or heard in the study area and Table 3 only records breeding data for species that breed in the study area.

Both refer to Fez pre-urban area. Table 3 is breeding population and Table 1 SM is the population all year round?

Response: Yes.

Table 3 – is - April-August correct or is it April – August? There are several lines with this

Response: Its April- August: Started in April and finished in August

Table 4: slightly edit the format with no deviation in the line

Response: Done

All in all the paper is well investigated and written. has good tension and describes the problem were well.

Reviewer 2 Report (Previous Reviewer 1)

This manuscript had been revised carefully according to the comments, and I have no further suggestions.

Author Response

Thank you 

Reviewer 3 Report (New Reviewer)

Title: Diversity of avian species in pre- urban landscapes surrounding Fez historical city Morocco): species richness, breeding populations and spatiotemporal evaluation of menacing factors

Please note the title has punctuation errors.

Authors: Wafae Squalli et al.

Journal: Diversity

Article ID: diversity-1962614

General overview

This study is an important contribution towards documenting the avian diversity of Morocco and in particular on the outskirts of the city of Fez. It is also a valuable contribution towards documenting the avian records of birds of North Africa that has, historically, received very little interest. However, the selection of study sites and why they were selected is lacking. It is also confusing that study sites held numerous habitat components – it would have been more appropriate to compare habitat types and their avian diversity with one another rather than block-based random study sites. This manuscript does require more extensive reworking to be suitable for publication.

Introduction

The introduction is clear enough on the uniqueness of the habitat and avian diversity to be explored. However, it is difficult to read through as there are numerous grammatical errors. In addition, and in the last paragraph of the introduction, objective 2 is the same as objective 1. Furthermore, and in objective 1, analyses of “breeding population” cannot fall under avian diversity. It is a completely separate investigation to determine if a population of a species is actually breeding or not – please refer to the hypothesis of “source and sink” populations. See: Sources, Sinks, and Population Regulation by H. Ronald Pulliam in The American Naturalist Vol. 132, No. 5 (Nov., 1988), pp. 652-661. As such, this should be a separate objective besides species richness and diversity.

As it stands, the introduction infers that the study is simply a checklist of avian species richness in a region surrounding an urban area and this has not sufficient scientific scope for publication. Rather, the introduction should highlight the importance of habitat diversity, water availability and possible human influence on the landscape and how this may impact on avian diversity. This is actually what the study was about.

Methods & data analyses

In the section discussing the study sites and under the definition of pre-urban “lands in the range of 30 km around the city as per-urban”, a large proportion of site A appears to fall directly within the city of Fez?

Also, in relation to the study sites selected, the authors need to justify more why these five sites where selected. Do these sites represent the diversity of habitat types surrounding the city? A bit more descriptive analyses of the sites is also required, e.g. what makes up the habitat characteristics of the “dump” site? etc…

The bird counting techniques seem appropriate as the habitats do differ to a large extent and count procedures are appropriately adapted for each site.

In terms of defining habitat characteristics and the possible negative influence of man-made structures (section 2.4, page 6 and Table 1), this is confusing and not well defined. I suggest it rather be more description and placed in the text and not as a table. Here, habitat characteristics should be clearly separated from human disturbance characteristics. Habitat characteristics should be defined as vegetation types and cover, dams and other water bodies and then the anthropogenic disturbances such as proximity to urban areas - I’m not sure how the latter can be defined as some study habitats are in urban areas, distance to city center makes no sense as you want to investigate the influence of urbanization and not the distance to the city center. Then include the other possible disturbance factors such as people and sewage tunnels – not shore how this is a disturbance either as many sewage plants and settling tanks attract wetland avifauna such as waders. The egg harvest description requires elaboration as this data seemed sourced from law enforcement. Livestock is also too vague, here you need precise heard size of the livestock species in question to calculate grazing capacity influence on the study site and compare them with other sites. This can all then be correlated to LANSAT imagery.

The statistical analyses seems appropriate. A Bray-Curstis similarity coefficient ordination comparing sites with one another based on avian community analyses would further compliment this manuscript.

Results

In this section, please make sure all scientific bird species names are italicized.

In line 182 it is mentioned that a total of 131 bird species were recorded. In line 191 it is mentioned that 85.5% were breeding species. Then, in line 204 the following is mentioned “In Total, 27 avian species were documented as breeding birds” 85.5% of 131 bird species is 112 species. This does not make sense.

Under 3.2, the authors will need to re-analyse the urban influence not to city center (as mentioned previously) but rather to first urban contact distance. In Table 4, what is mean by “populations”? In the caption to Table 4, diversity is not determined by number of species (species richness) but by a combination of species richness and number of individuals of each species. What statistical procedure was used to calculate this outcome presented in this table?

With regards to Figure 4, this figure does not represent the expansion of 20 years ago and the current expansion, it only represents the current condition. The authors may want to consider a map 20 years back and a current map for comparison as the reader has no idea of the expansion rate over 2 decades.

Discussion

The discussion is adequate.

Editorial suggestions

Line 42, page 1: Earth is a proper noun, hence it must be spelt with a capital “E”. On the same line, remove the -“- before citation number

Line 43, page 1: replace “greatest” with “largest”. The same in line 44.

Lines 44 and 45: with regards this sentence “Morocco houses more than 31,000 living species of which about 11 % are endemic” I’m not sure how accurate this is as it has not citation. Bacteria and other microbes are also living species.

Line 45, page 2: replace “avifauna class” with “avifauna”

Line 45 and 46: What does this mean “Morocco holds nearly half thousands of birds” ?

Line 48: thirty-eight should not be hyphenated

Please note: Table SM 1 – “UICN” should be “IUCN”

I HAVE NOW STOPPED AT THIS POINT MAKING EDITIORIAL SUGGESTIONS AS THEY ARE TOO FREQUENT AND TOO MANY. THE AUTHORS WILL NEED TO SUBMIT THE MANUSCRIPT TO AN ENGLISH EDITOR TO BRING IT UP TO PUBLICATION QUALITY.

Author Response

General overview

This study is an important contribution towards documenting the avian diversity of Morocco and in particular on the outskirts of the city of Fez. It is also a valuable contribution towards documenting the avian records of birds of North Africa that has, historically, received very little interest. However, the selection of study sites and why they were selected is lacking. It is also confusing that study sites held numerous habitat components – it would have been more appropriate to compare habitat types and their avian diversity with one another rather than block-based random study sites. This manuscript does require more extensive reworking to be suitable for publication.

Introduction

The introduction is clear enough on the uniqueness of the habitat and avian diversity to be explored. However, it is difficult to read through as there are numerous grammatical errors. In addition, and in the last paragraph of the introduction, objective 2 is the same as objective 1. Furthermore, and in objective 1, analyses of “breeding population” cannot fall under avian diversity. It is a completely separate investigation to determine if a population of a species is actually breeding or not – please refer to the hypothesis of “source and sink” populations. See: Sources, Sinks, and Population Regulation by H. Ronald Pulliam in The American Naturalist Vol. 132, No. 5 (Nov., 1988), pp. 652-661. As such, this should be a separate objective besides species richness and diversity.

As it stands, the introduction infers that the study is simply a checklist of avian species richness in a region surrounding an urban area and this has not sufficient scientific scope for publication. Rather, the introduction should highlight the importance of habitat diversity, water availability and possible human influence on the landscape and how this may impact on avian diversity. This is actually what the study was about.

Response: Our objective is to explore (first) then show the importance of habitats and how human factors influence birds. Therefore we didn’t modify completely the introduction, but we have added an entire paragraph as recommended.

It has been frequently claimed that per-urban areas, including reservoirs, greenbelts, and remaining natural habitats, support a substantial avian species assemblage [6,7]. For instance, inside the administrative boundaries of Prague city, 127 of the 199 avian species that breed in the Czech Republic can be found [8]. However, this diversity is impacted by anthropogenic factors, including urbanization, pollution, traffic, degradation of natural ecosystems, noise and disturbances [9,10]. While these features are reasonably well stud-ied north of the Mediterranean, data are not yet available from North Africa, despite the importance of this area for both local and migratory birds [10–12].

Methods & data analyses

In the section discussing the study sites and under the definition of pre-urban “lands in the range of 30 km around the city as per-urban”, a large proportion of site A appears to fall directly within the city of Fez?

Response: These are per-urban based on master plan of the city created during 2000. The expansion of urbanization during the last 20 years makes Oued Fez, El Gaada dam, and Ain Chekef nearly 100% urbanized.

Also, in relation to the study sites selected, the authors need to justify more why these five sites where selected. Do these sites represent the diversity of habitat types surrounding the city? A bit more descriptive analyses of the sites is also required, e.g. what makes up the habitat characteristics of the “dump” site? etc…

Response: We selected dominant ecosystems in the area, Farmlands and dump (Ain Lbida), reservoirs ( EL Mahraz and El Gaada), Forest (Ain Chekef), River (Oued Fez). Equally, these sites represent the diversity of ecosystems in the area.

EX: Ain Lbida dump offers food and habitats for wintering birds when foraging resources are short in winter. It combine between farmlands and wasts which is rare in Morocco.

Introduction

We selected the Fez region because of its central location in Morocco which is close to the humid Atlas Mountains, offering a wide range of ecosystems including forests, reservoirs, and farmlands.

The study area offers a last foraging opportunity for long-distance migratory birds be-fore they cross the Sahara [26]. Equally, the region holds several wetlands, considered as important RAMSAR sites for water birds. Generally, this study was designed to explore the importance of habitats for birds and to characterize the most threatening factors to both habitats and birds.

Methodes

These selected sites are the dominant ecosystems in the area and represent the only systems located in the per-urban zones between Fez city and rural farmlands of Saiss plain. Equally, these ecosystems are the only sites in which we can investigate the interaction between avian communities, habitats features, and anthropogenic activities.

The bird counting techniques seem appropriate as the habitats do differ to a large extent and count procedures are appropriately adapted for each site.

In terms of defining habitat characteristics and the possible negative influence of man-made structures (section 2.4, page 6 and Table 1), this is confusing and not well defined. I suggest it rather be more description and placed in the text and not as a table. Here, habitat characteristics should be clearly separated from human disturbance characteristics. Habitat characteristics should be defined as vegetation types and cover, dams and other water bodies and then the anthropogenic disturbances such as proximity to urban areas - I’m not sure how the latter can be defined as some study habitats are in urban areas, distance to city center makes no sense as you want to investigate the influence of urbanization and not the distance to the city center. Then include the other possible disturbance factors such as people and sewage tunnels – not shore how this is a disturbance either as many sewage plants and settling tanks attract wetland avifauna such as waders. The egg harvest description requires elaboration as this data seemed sourced from law enforcement. Livestock is also too vague, here you need precise heard size of the livestock species in question to calculate grazing capacity influence on the study site and compare them with other sites. This can all then be correlated to LANSAT imagery.

Response: This suggestion is in contradiction with the comment of the first reviewer who recommended to prepare these variables in table as presented. But if the Editor insist, I’m ready to rebuild it as you recommend.

The statistical analyses seem appropriate. A Bray-Curstis similarity coefficient ordination comparing sites with one another based on avian community analyses would further compliment this manuscript.

Response:  We realized the test as recommended. And we included the findings in the results

Figure: Bray-Curstis similarity coefficient ordination comparing studied sites based on avian communities

Results

In this section, please make sure all scientific bird species names are italicized.

Response: Done

In line 182 it is mentioned that a total of 131 bird species were recorded. In line 191 it is mentioned that 85.5% were breeding species. Then, in line 204 the following is mentioned “In Total, 27 avian species were documented as breeding birds” 85.5% of 131 bird species is 112 species. This does not make sense.

Response: the status “Breeding” means that the species breeds in Morocco, and among the 112 recorded species we have approved 27 species that breed in the study area

Under 3.2, the authors will need to re-analyse the urban influence not to city center (as mentioned previously) but rather to first urban contact distance. In Table 4, what is mean by “populations”? In the caption to Table 4, diversity is not determined by number of species (species richness) but by a combination of species richness and number of individuals of each species. What statistical procedure was used to calculate this outcome presented in this table?

Response: We reanalyzed all predictors and we added the distance to the nearest urban zone as suggested by the reviewer.

We evaluated the relation between the number of species, their populations, and habitat characteristics (area, distance to nearest urban zone, and distance to urban center) via linear regression. Bird species and population size were considered dependent variables, while habitat area, distance to nearest urban zone, and distance to the urban center were considered predictors.

With regards to Figure 4, this figure does not represent the expansion of 20 years ago and the current expansion, it only represents the current condition. The authors may want to consider a map 20 years back and a current map for comparison as the reader has no idea of the expansion rate over 2 decades.

Response: We have rebuilt new figure as recommended.

Discussion

The discussion is adequate.

Editorial suggestions

Line 42, page 1: Earth is a proper noun, hence it must be spelt with a capital “E”. On the same line, remove the -“- before citation number

Response: Done

Line 43, page 1: replace “greatest” with “largest”. The same in line 44.

Response: Done

Lines 44 and 45: with regards this sentence “Morocco houses more than 31,000 living species of which about 11 % are endemic” I’m not sure how accurate this is as it has not citation. Bacteria and other microbes are also living species.

Respone: See these references:

M Menioui. Biological Diversity in Morocco - Global Biodiversity, 2018

Taybi, A. F., Glöer, P. & Mabrouki, Y.: Two new species of Mercuria Boeters, 1971 from Morocco (Gastropoda, Hydrobiidae). Nat. Croat., Vol. 31, No. 1, 63-69, Zagreb, 2022.

Line 45, page 2: replace “avifauna class” with “avifauna”

Response: Done

Line 45 and 46: What does this mean “Morocco holds nearly half thousands of birds” ?

Response: 500 different species of migratory, breeding, and wintering birds was found in Morocco.

Line 48: thirty-eight should not be hyphenated

Response: Done

Please note: Table SM 1 – “UICN” should be “IUCN”

Response: Done

I HAVE NOW STOPPED AT THIS POINT MAKING EDITIORIAL SUGGESTIONS AS THEY ARE TOO FREQUENT AND TOO MANY. THE AUTHORS WILL NEED TO SUBMIT THE MANUSCRIPT TO AN ENGLISH EDITOR TO BRING IT UP TO PUBLICATION QUALITY.

Response : The English is improved now by the Editor Michael Wink

Round 2

Reviewer 3 Report (New Reviewer)

Please see uploaded PDF file.

Author Response

This manuscript is a resubmission of an earlier submission. The following is a list of the peer review reports and author responses from that submission.

Round 1

Reviewer 1 Report

This manuscript is an interesting report on the relationship between the factors and the birds in city. However, there were some flaws should be addressed by the authors:

1. How to define "pre-urban"? In my opinion, it was very difficult to define it given that the long history of human disturbance in this area.

2. The spatial pattern of the study sites should be considered, as the distance between each other and the surrounding habitat should be important to the birds in these areas. By the way, the developing history of each study site should be considered.

3. I also mentioned some minor concerns here:

1) Line 35-39: It is not the reality, as there were many reports in the past.

2) Line 41: " should be added to the end of “the earth”.

3) Line 43: "31.000" should be "31,000".

4) Line 83: "FiFigure 1" should be deleted.

5) Line 92: "Point-counts" and "Point counts" should be unified in a manuscript.

6) The duration at a point was different, and it was confused for us.

7) Figure 2: It was important to show the spatial pattern of these sites.

8) Is there only a urban centre toursits in the city or in a study site?

9) Liveostock: it was not clear of "size of whole animals recorded for each site". The body size? or the population size?

Reviewer 2 Report

The paper is well investigated and written. In my opinion it needs only minor corrections.

Introduction has good tension and describes the problem were well. I am fin with it.

Material and methods:

Text figure 1 delete FiFigure 1.

In figure 2 the authors provide an overview of the study sites, I would prefer to have a map to so the location of the study sites in relation to each other.

Table 1 needs another format, otherwise it is impossible to read. In my opinion the column “How it is expected to impact birds” is not necessary.

Line 122 livestock? Livestock numbers?

In my opinion it is not necessary to describe the used ecological indices in detail. A description what index describes what and a reference should be enough and if somebody is interested how to calculate it is possible to look in the reference.

It maybe a more theoretical problem but to sum up individuals recorded by different survey methods could be a problem which should be addressed a least in the method section.

Please define status accidental, breeding, migrating, wintering, summering

Results

Line 178-180 I understand what the authors try say but reducing all recorded species to passerines, water birds and raptors maybe misleading especially water birds is a quite unspecific term. Afterwards the authors focus a families etc. therefor I would skip the sentence.

193 phonological – phenological

Table 2 – Abundance should be named population size. In the method section the authors describe the survey of the individuals and how they estimate population size but in table 2 they present an estimation with quite range. Is the number a estimation are does it show the variation over the years?

Maybe table 2 should be converted to an appendix.

In line 179 the authors refer to 131 species found and in line 206 they talk about 164 species. How does this fit together?

Table 3: what means 0 in the upper left corner. Use Taxa, Shannon-Weaver etc. instead of Taxa_S, Shannon_H etc.

Table 4: Is the breeding period important?

Table 5: Is it necessary to present the values for residuals and total (corr) in the table? The line area for population is double

Figure 6: Legends a hard to read because the are turned 90°. Try to have one orientation of the legend and the enumeration. Start with A on the top.

Discussion and conclusions are okay for me.

Reviewer 3 Report

I read the manuscript with interest. It provides information about the avifauna in an urban area in Morocco. In fact the manuscript is more or less well constructed, however, the introduction part could be written a bit better, focusing on earlier studies similar to this one. Information about Morocco and the study area should be moved to Material and methods. Two years of field work for such type of a study is quite a short period in my opinion. I feel that these results are more interesting for a more regional journal dealing with birds. The scientific merit of the paper is low, therefore I cannot recommend it to the journal Diversity.